# Radiation Therapy for Stage IIA/B Seminoma: Modeling Secondary Cancer Risk for Protons and VMAT versus 3D Photons

**DOI:** 10.3390/cancers16040784

**Published:** 2024-02-15

**Authors:** Jennifer Pursley, Kyla Remillard, Nicolas Depauw, Grace Lee, Clemens Grassberger, Harald Paganetti, Jason A. Efstathiou, Sophia C. Kamran

**Affiliations:** 1Department of Radiation Oncology, Massachusetts General Hospital, Harvard Medical School, Boston, MA 02114, USAskamran@mgh.harvard.edu (S.C.K.); 2Department of Radiation Oncology, University of Washington Medical Center, Seattle, WA 98195, USA

**Keywords:** testicular seminoma, proton therapy, secondary cancer modeling

## Abstract

**Simple Summary:**

Radiation therapy is an effective treatment for stage IIA and select stage IIB testicular seminoma patients. However, given the young age and long life expectancy of these patients, there are concerns about the risk of secondary cancers from radiation therapy. This study assessed the expected differences in secondary cancer risk for stage II seminoma patients receiving proton pencil-beam scanning (PBS) and photon VMAT compared to 3D conformal photon treatment plans. For 10 seminoma patients treated at our institution, 3D, VMAT, and PBS plans were generated. The percentage increase in the lifetime risk of developing cancers secondary to the radiation was modeled for each plan. On average, PBS reduced the mean radiation dose to critical organs by 60% and reduced the increase in lifetime secondary cancer risk by 55% compared to photons. Proton therapy should be considered for this patient population as it shows the potential to reduce late side effects.

**Abstract:**

Radiation therapy (RT) is an effective treatment for stage IIA and select stage IIB seminomas. However, given the long life expectancy of seminoma patients, there are concerns about the risk of secondary cancers from RT. This study assessed differences in secondary cancer risk for stage II seminoma patients following proton pencil-beam scanning (PBS) and photon VMAT, compared to 3D conformal photon RT. Ten seminoma patients, five with a IIA staging who received 30 GyRBE and five with a IIB staging who received 36 GyRBE, had three RT plans generated. Doses to organs at risk (OAR) were evaluated, and secondary cancer risks were calculated as the Excess Absolute Risk (EAR) and Lifetime Attributable Risk (LAR). PBS reduced the mean OAR dose by 60% on average compared to 3D, and reduced the EAR and LAR for all OAR, with the greatest reductions seen for the bowel, liver, and stomach. VMAT reduced high doses but increased the low-dose bath, leading to an increased EAR and LAR for some OAR. PBS provided superior dosimetric sparing of OAR compared to 3D and VMAT in stage II seminoma cases, with models demonstrating that this may reduce secondary cancer risk. Therefore, proton therapy shows the potential to reduce acute and late side effects of RT for this population.

## 1. Introduction

The incidence rate of testicular cancer in the US has been increasing for several decades, with an average annual rate of 5.7 per 100,000 men in the years 2014–2018 [1,2]. The increase is mostly in seminoma, which is diagnosed at an average age of about 33. Seminoma remains a highly curable disease, with an average annual death rate of less than 0.3 per 100,000 [1], so the risk of treatment-related morbidity is one of the principal concerns influencing treatment management decisions.

Most testicular seminoma patients (~85%) are diagnosed with clinical stage I (CS1) disease, while ~10% of new cases consist of clinical stage IIA (CS2A) or IIB (CS2B) disease, defined by metastatic retroperitoneal lymph nodes which are up to 5 cm in diameter [3,4]. Treatment options for CS2A/B disease consist of radiation therapy (RT) or cisplatin-based chemotherapy, with RT preferred for CS2A and select CS2B disease [5,6]. Modern 3D conformal RT (3D) uses two opposed fields, anterior-to-posterior (AP) and posterior-to-anterior (PA), to provide a uniform dose coverage of the para-aortic and ipsilateral pelvic nodes [7]. Nodal volumes are contoured on a CT scan and enlarged lymph nodes are targeted and boosted to a higher dose [8].

Despite the low doses of radiation which are used for CS2A/B seminoma, there are documented acute and late side effects, which are related to the volume of normal tissue irradiated and the dose received [9]. Acute side effects from pelvic fields include fatigue, decreased blood cell counts, and mild gastrointestinal distress (nausea, diarrhea). While these acute side effects are generally mild and manageable, reducing the volume of healthy tissue and critical organs at risk (OAR) irradiated could further reduce their impact. Late side effects, occurring more than 5 years after radiation treatment, are generally considered minimal following low doses of radiation, with the largest concern being secondary radiation-induced cancer [10]. Reducing the volume of healthy tissue and OAR irradiated is also expected to reduce the risk of secondary cancer. 

With contoured target volumes, intensity-modulated RT deliveries such as IMRT and VMAT are also possible [11,12]. However, these techniques increase the volume of irradiated healthy tissue compared to 3D, therefore, potentially increasing the risk of secondary cancers [13]; a major concern for this young population. Proton therapy has also been explored for seminoma patients, as protons deliver the maximum dose at the end of their range (i.e., the Bragg Peak) and the dose rapidly falls off after this peak, limiting the low dose delivered to many OAR [14,15,16,17]. A prior modeling analysis demonstrated that the use of protons for clinical stage I (CS1) seminoma could theoretically avert 300 excess secondary malignancies among 10,000 men treated at age 39 and surviving to age 75 [15]. A recent study reported on control and toxicity outcomes for CS1-2B seminoma patients who were treated with proton therapy and performed a dosimetric comparison with photons [18].

In this study, we compare the dose to relevant OAR for CS2A/B seminoma patients and model the rates of radiation-induced secondary cancers to compare the expected morbidity from three forms of radiation treatment: 3D photons, VMAT photons, and proton pencil-beam scanning (PBS). Each radiation option delivers the same prescription dose to the target but delivers different amounts of radiation to the OAR, which can lead to differences in both acute and late radiation-induced toxicities.

## 2. Materials and Methods

After receiving institutional review board approval for this retrospective study, the ten most recent patients treated for CS2 seminoma (five with stage CS2A and five with CS2B) at Massachusetts General Hospital from 2011 to 2021 were selected; patient characteristics are provided in Table 1. The mean age was 40 (median 39, range 32–54 years). All patients received a helical CT scan with a 2.5 mm slice thickness for planning. Five patients were clinically treated with protons and five with 3D photons. The majority of the involved lymph nodes treated with a higher-dose boost were in the para-aortic region. All patients had additional organ structures contoured by a radiation oncologist to achieve a consistent set of 13 organs for the dosimetric analysis: the bladder, large bowel, small bowel, spinal cord, kidneys, liver, pancreas, rectum, stomach, femurs, pelvic ilium, sacrum, and vertebral bodies from T10 to L5.

Treatment plans were generated for each patient for three modalities (3D, VMAT, and PBS) by two experienced treatment planners in accordance with institutional clinical standards. The clinical target volume (CTV) for the initial treatment of 20 GyRBE in 10 fractions was created around the aorta and inferior vena cava from the T11/T12 interface to L5, including the ipsilateral hemipelvis to the top of the acetabulum, and adding the CTV around any enlarged lymph nodes. A 5 mm uniform expansion was applied to generate the planning target volume (PTV) for photon plans. Enlarged lymph nodes, generally with a 1–2 cm uniform CTV expansion and an additional 5 mm uniform PTV expansion, received a sequential boost of 10 GyRBE in five fractions for CS2A or 16 GyRBE in eight fractions for CS2B patients.

PBS proton plans were generated in the Astroid treatment planning system (.decimal, LLC, Sanford, FL, USA) for a ProTom Radiance 330 spot-scanning system. The PBS plans used a single PA field to reduce the range uncertainty from treating AP through luminal, air-filled abdominal organs. No PTV expansions were used as the PTV expansion is not an appropriate surrogate for treatment robustness for proton planning. Instead, plans were generated per the clinical standard of ensuring that target coverage remained adequate under 5 mm setup shifts and 3.5% range uncertainties. A constant relative biological effectiveness (RBE) value of 1.1 was assumed for proton doses [19]. Photon plans were generated in RayStation 10A (RaySearch Laboratories, Stockholm, Sweden) for an Elekta Agility linear accelerator with 160 multi-leaf collimators. The 3D AP-PA 10 MV photon plans were generated with a 0.7 cm expansion from the PTV to the lateral block edges and a 1 cm expansion to the superior and inferior block edges, with subfields used to reduce hot spots and improve dose uniformity. The VMAT 6 MV photon plans were generated using two 358° arcs and optimized to provide a conformal high-dose distribution around the PTV, with maximum dose constraints on the bowel and spinal cord, and low-dose constraints on the kidneys. 

Treatment plans were not normalized to the same target coverage; rather, each plan was generated to the best clinically acceptable coverage available for that modality. Generally, for 3D plans, 95–100% of the PTV and 99–100% of the CTV were covered by 95% of the prescription dose. For VMAT, generally 95% of the PTV and 100% of the CTV were covered by the prescription dose. For PBS, 100% of the CTV was covered by the prescription dose.

Dosimetric comparisons between treatment plans were made using both the mean dose and a generalized Equivalent Uniform Dose (EUD) [20]. The EUD represents the uniform dose that would produce the same risk of critical-organ morbidity as the actual inhomogeneous dose distribution. This allows for a more consistent method of comparing the potential for acute treatment-related morbidity than the mean physical dose received by an organ. The model requires an organ-specific “a” parameter to compute the EUD from the dose–volume histogram; the values used for organs in this study are shown in Table 2 and are consistent with a previous study on CS1 seminoma [14].

The risks of developing a secondary cancer were modeled using the Organ Equivalent Dose (OED) [21,22,23,24,25], which uses tissue-dependent parameters from the linear–quadratic model of cell kill to account for the non-uniform dose distributions delivered during radiotherapy. The OED replaces the homogeneous dose of classical models used to estimate the risk of secondary cancer, which are based on atomic bomb survivors [26,27,28]. The methodology used in this study has been described in a prior secondary cancer analysis for breast RT [29] and assumes an α/β ratio of 3 for all normal organs. Secondary cancer risks are expressed as the Excess Absolute Risk (EAR), reported as the increase in cancers per 10,000 person–years. The EAR is used to model the Lifetime Attributable Risk (LAR), which is the increase in the likelihood of developing a malignancy over a person’s lifetime. In this study, the EAR and LAR were projected from the age at treatment to age 70.

The EAR and LAR were calculated using the organ-specific parameters and models proposed by Schneider et al. [25], which were based on an analysis of Hodgkin’s lymphoma patients. While using these values could lead to an overestimation of the absolute risk due to the genetic susceptibility of Hodgkin’s patients to malignancies, the comparison of relative risk between different treatment plans should not be affected. Organ-specific parameters for the kidneys and pancreas were not available from Schneider’s work. These organs were assigned the same parameters as the small bowel based on the work of Preston et al. [26], who found similar secondary cancer induction coefficients for the pancreas, kidneys, and small bowel in an analysis of atomic bomb survivors. 

Results for the different modalities were compared using a Wilcoxon signed-rank test with *p* < 0.05 indicating statistical significance; statistical analysis was done using R [30].

## 3. Results

Figure 1 shows a comparison of the three planning modalities for a single representative patient case. The VMAT and PBS plans demonstrate high conformality of the prescription isodose lines (IDL) to the target volumes, while the 3D AP-PA plan shows the prescription IDL encompassing most of the irradiated volume. On average, the ratio of the V20 (volume encompassed by the 20 GyRBE IDL) for VMAT compared to 3D was 0.36 ± 0.04 (95% CI; *p* < 0.0001); the ratio was similar for PBS compared to 3D at 0.33 ± 0.05 (95% CI; *p* < 0.0001). The mean dose to the body was 4.2 ± 0.5 Gy for 3D, 4.3 ± 0.6 Gy for VMAT, and 1.8 ± 0.2 Gy for PBS; only PBS significantly reduced the mean dose compared to 3D (*p* < 0.0001).

Comparisons of CS2A and CS2B patients showed identical trends in the dosimetric and secondary cancer analyses, so results from all 10 patients were combined.

### 3.1. Dosimetric Comparison

Table 2 shows a comparison of the EUDs and mean doses for the 13 contoured OAR averaged over 10 patients. The average mean doses over the three modalities are also displayed in Figure 2. Both the EUDs and mean doses for all OAR were significantly reduced by PBS compared to 3D. Compared to 3D, VMAT significantly increased the mean dose for several organs (the bladder, large bowel, liver, and rectum) although the EUD only significantly increased for the liver, from 2.9 GyRBE to 4.5 GyRBE on average. Other organs saw a significant decrease in the mean dose and EUD for VMAT compared to 3D. Compared to PBS, the spinal cord was the only organ for which the mean dose and EUD were significantly reduced by VMAT; all other organ doses were similar or significantly improved by PBS.

### 3.2. Secondary Cancer Risk Modeling

Table 3 summarizes the organ-specific secondary cancer risk per 10,000 person–years relative to baseline for each treatment modality, projected to 70 years of age and averaged over 10 seminoma patients. The risk of developing leukemia, due principally from dose to active bone marrow, was not included in this study because of the lack of validated models. As a substitute, we compared V10 for the pelvic ilium across the three modalities; averaged over the 10 patients, the V10 was significantly reduced by PBS compared to either 3D or VMAT (*p* < 0.002; V10 = 98 ± 34, 103 ± 47, and 65 ± 34 cc for 3D, VMAT, and PBS respectively). The EARs and LARs were significantly reduced for PBS compared to 3D photons for all OAR. Compared to 3D, VMAT significantly increased the EARs and LARs for the liver, stomach, and rectum, but did not significantly change them for the bladder and kidneys, and significantly reduced them for the spinal cord, large and small bowel, and pancreas. Compared to VMAT, PBS gave similar EARs and LARs for the spinal cord and pancreas, but significantly reduced them for all other OAR.

When combining all OAR, the total average EARs were 29.1, 26.7, and 12.3, and the total average LARs were 3.4%, 3.2%, and 1.52% for 3D, VMAT, and PBS, respectively. Therefore, compared to 3D, VMAT reduced the total EAR by 8% and LAR by 6%, while PBS reduced the total EAR by 58% and LAR by 55%. This indicates that treating CS2A/B seminoma with proton PBS rather than 3D photons may lead to a greater than two-fold reduction in secondary cancers.

## 4. Discussion

Radiation therapy is an effective treatment for early-stage testicular seminoma, and only relatively low doses are required. However, as this patient population is young, with a high cure rate and a long life expectancy, the late side effects of treatment are a major consideration in disease management. A prior analysis indicated that the use of double-scatter protons in CS1 seminoma limited the dose to normal structures and reduced the predicted secondary cancer risk compared to 3D photon irradiation [14]. In this study for CS2A/B seminoma, we demonstrated that the use of PBS protons significantly reduced the dose to OAR compared to both 3D and VMAT irradiation. This dosimetric sparing translated into decreased estimated risks of secondary cancer, with a greater than two-fold reduction for PBS compared to 3D. VMAT showed dosimetric sparing of some organs compared to 3D, but the EARs and LARs increased for several organs (liver, stomach, and rectum) and had little reduction for other organs. Thus, in considering potential acute and late side effects, this study does not find a benefit to VMAT over 3D while a significant benefit is expected from PBS.

While the results of this study were strongly in favor of PBS, this was a retrospective modeling study and there were no patient outcomes to compare. A recent retrospective study reported favorable cancer control and toxicity outcomes in 24 CS1-2B seminoma patients who were treated with protons and predicted the risk of secondary malignancies using two different models [18]. A case report described the efficacy and advantages of proton RT for seminoma [31], and a single institution report detailed the outcomes of four stage I seminoma patients treated with proton RT [32]. There have been database studies showing lower rates of secondary malignancies between patients treated with protons versus photon 3D or IMRT [33,34,35], so there is support for secondary cancer modeling predictions in patient outcome data. However, the clinical impact of dosimetric improvements would be best measured by a prospective treatment protocol that assesses outcomes and toxicity endpoints.

There are large uncertainties in the model parameters used to estimate secondary cancer risks, which are not included in the secondary cancer estimates in Table 3. However, since this study is a comparison of secondary cancer risks based on dosimetric changes from different modalities, we do not expect the comparisons to be significantly impacted by modifications to the model parameters. We also did not consider the effects of a low dose from scatter radiation or secondary neutrons outside of the radiation field, and the potential for secondary cancers in out-of-field organs. For PBS, it has been shown that secondary radiation from neutrons is negligible [36]. The out-of-field secondary cancer risk of VMAT is estimated to be up to 50% higher than 3D, which is another reason to prefer 3D over VMAT for seminoma patients [37].

One limitation of this study is that results could be impacted by institutional guidelines and planning techniques, particularly for VMAT. VMAT plans were optimized to meet physical dose constraints, with particular attention given to the conformity of the prescription IDL to the targets and sparing low doses to the kidneys. If the optimizations were performed based on the biological dose, with more consideration of the EUD than the mean or maximum dose, then the dosimetric and secondary cancer analysis results for VMAT compared to 3D and PBS could change. This study also assumed a constant proton RBE of 1.1, consistent with current clinical practice. However, the RBE is known to depend on physical and biological factors that vary over the proton range, and the RBE could be higher than 1.1 in some regions which could affect the dose and secondary cancer risks in OAR [38]. It may be worth revisiting the comparison of these three modalities in a future study when there are validated models for biological dose optimization.

Another limitation of this study is the small sample size of ten CS2 patients, with ages ranging from 32 to 54, and large variability in the anatomic location of the enlarged lymph nodes which received an additional 10 GyRBE (for CS2A) or 16 GyRBE (for CS2B). This anatomic variability resulted in large variations in the mean doses and EUDs to some organs across patients, as the mean organ dose increased if the boost target was located nearby. However, the Wilcoxon signed-rank test is designed to compare paired samples without relying on a normal distribution around a mean value, so this variability should have a minimal impact on the results and corresponds to a realistic patient population.

## 5. Conclusions

This study compared three radiation treatment modalities for ten clinical stage II seminoma patients and found that PBS proton therapy significantly reduced the physical dose and the predicted secondary cancer risk for all patients compared to 3D and VMAT photons, potentially resulting in reduced acute and late side effects for this patient population. This is consistent with the findings of a previous study in stage I seminoma [14]. Access to proton therapy is still relatively limited in the United States [39], so PBS is considered a limited resource given its expense and lack of availability. However, proton therapy has shown indisputable benefits in pediatric cancer, where patients are treated at a young age and have a high cure rate with a long life expectancy; patients with seminoma also have a young average age at diagnosis and high curability and could be expected to similarly benefit from proton therapy, yet insurance coverage for proton therapy is routinely denied for this population. ASTRO Model Policies for proton-beam treatment categorizes testicular cancer into group two, for which coverage is recommended only if sparing of the surrounding healthy tissue cannot be adequately achieved with photon-based radiotherapy [40]. This policy may need to be revisited in light of these studies demonstrating a benefit, with a reduced predicted secondary malignancy risk using protons to re-categorize testicular seminoma as group one (for which health insurance coverage is recommended). Future studies should undertake prospective treatment protocols to fully quantify the clinical benefits of proton therapy in seminoma treatment in regard to acute and late side effects and patient-reported quality-of-life outcomes.

## Figures and Tables

**Figure 1 cancers-16-00784-f001:**
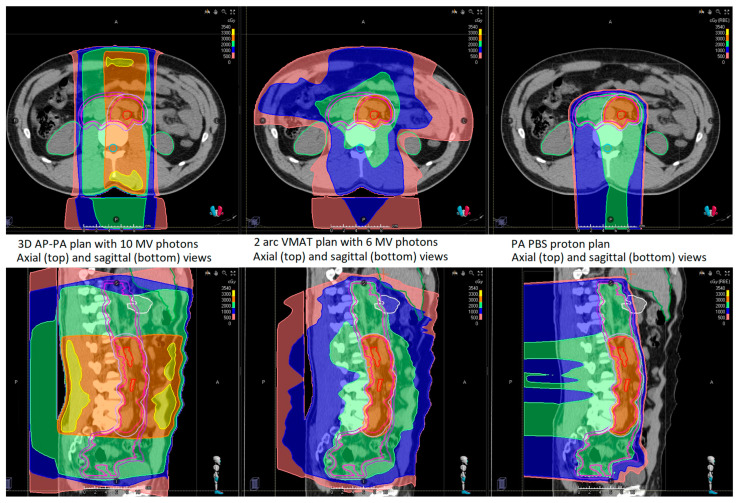
Axial (top row) and sagittal (bottom row) views of a representative stage IIA seminoma patient receiving 20 GyRBE to the initial CTV and a 5 mm PTV expansion for photons, with an additional 10 GyRBE boost to the enlarged lymph nodes with a 2 cm CTV expansion, and a 5 mm PTV expansion for photons. From left to right, the plans shown are 3D with 10 MV opposed photon fields, VMAT with two 6 MV photon arcs, and proton PBS with one posterior field.

**Figure 2 cancers-16-00784-f002:**
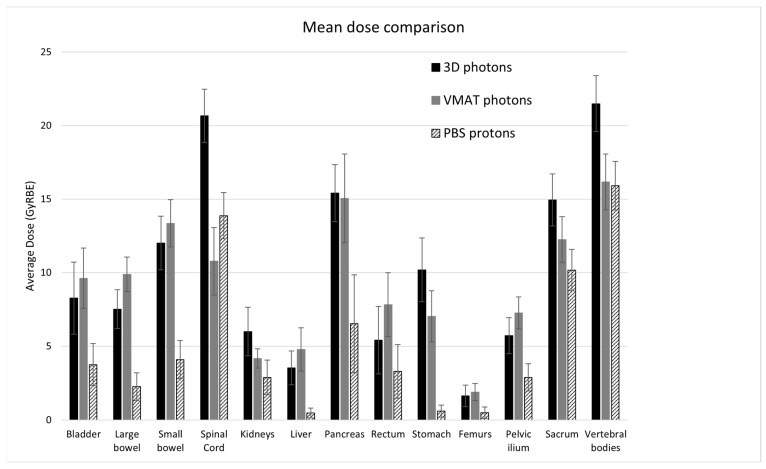
Comparison of the mean dose for 13 organs averaged over ten CS2 seminoma patients planned with three modalities. Error bars represent the 95% confidence intervals for the mean.

**Table 1 cancers-16-00784-t001:** Patient characteristics. The treatment modality shown in the table is the one used clinically.

Clinical Stage	Age at Treatment	Treatment Modality	Laterality	Number of LN	Largest LN Dimension (cm)	LN Location
IIA	41	PBS	Left	2	1.4	Left common iliac, level L4/L5; left para-aortic, level L2/L3
IIA	38	PBS	Left	4	1.0	Left para-aortic, level L3; left external iliac, levels L2 and L5/S1 and S5
IIA	48	PBS	Right	1	2.0	Anterior to right psoas, level L5/S1
IIA	38	PBS	Left	>5	1.0	Left para-aortic, top of L2 to bottom L3 and level L4
IIA	41	3D	Left	1	1.2	Left para-aortic, level L2/L3
IIB	41	PBS	Right	>5	2.3	Aortocaval, level L1/L2; multiple, level L3
IIB	32	3D	Right	1	2.7	Right external iliac, levels S3–S5
IIB	54	3D	Right	1	3.8	Below the renal veins, level L3
IIB	38	3D	Left	1	2.1	Left para-aortic, level L1/L2
IIB	33	3D	Left	2	3.3	Left para-aortic, level mid-L2; anterior to left psoas, level L4/L5

Abbreviations: 3D = 3D conformal photons; PBS = pencil-beam scanning protons; LN = lymph nodes.

**Table 2 cancers-16-00784-t002:** Equivalent Uniform Dose (EUD) and mean dose values to critical normal structures averaged over 10 patients with 95% confidence intervals for the mean given in parentheses. The “a” value is an organ-specific parameter used to compute the EUD from a non-uniform dose distribution. All PBS proton EUD and mean dose values were significantly lower than 3D photon values (*p* < 0.05). Values that are significantly different between 3D and VMAT are labeled with * while values that are significantly different between VMAT and PBS are labeled with ^†^.

Organs	a	EUD 3D (GyRBE)	EUD VMAT (GyRBE)	EUD PBS (GyRBE)	Mean Dose 3D (GyRBE)	Mean Dose VMAT (GyRBE)	Mean Dose PBS (GyRBE)
Bladder	2	11.3 (8.8, 13.8)	11.1 (9.0, 13.2)	7.4 (5.4, 9.5) ^†^	8.3 (5.8, 10.7)	9.6 (7.6, 11.7) *	3.8 (2.3, 5.2) ^†^
Large bowel	5	19.4 (18.0, 20.8)	15.5 (14.0, 17.0) *	13.1 (11.0, 15.2) ^†^	7.5 (6.2, 8.8)	9.9 (8.7, 11.1) *	2.3 (1.3, 3.2) ^†^
Small bowel	5	22.4 (20.3, 24.4)	19.4 (17.6, 21.2) *	16.8 (14.8, 18.8) ^†^	12.0 (10.2, 13.8)	13.4 (11.7, 15.0)	4.1 (2.8, 5.4) ^†^
Spinal cord	20	27.7 (24.9, 30.5)	18.0 (15.0, 20.9) *	20.1 (17.4, 22.9) ^†^	20.7 (18.8, 22.5)	10.8 (8.5, 13.1) *	13.9 (12.3, 15.4) ^†^
Kidneys	1.5	7.6 (5.7, 9.5)	4.5 (3.7, 5.2) *	4.5 (3.1, 6.1)	6.0 (4.3, 7.6)	4.2 (3.5, 4.8) *	2.9 (1.7, 4.1) ^†^
Liver	0.8	2.9 (1.9, 3.9)	4.5 (3.1, 5.9) *	0.25 (0.05, 0.45) ^†^	3.5 (2.4, 4.7)	4.8 (3.3, 6.2) *	0.48 (0.15, 0.81) ^†^
Pancreas	5	20.4 (18.0, 22.8)	17.7 (14.7, 20.8) *	14.7 (11.4, 18.0) ^†^	15.4 (13.5, 17.4)	15.1 (12.0, 18.1)	6.5 (3.2, 9.9) ^†^
Rectum	5	11.9 (9.4, 14.4)	11.3 (8.8, 13.8)	10.0 (7.4, 12.6) ^†^	5.4 (3.1, 7.7)	7.8 (5.6, 10.0) *	3.3 (1.5, 5.1) ^†^
Stomach	7	19.9 (17.2, 22.7)	11.1 (8.8, 13.3) *	7.8 (5.3, 10.3) ^†^	10.2 (8.0, 12.4)	7.0 (5.3, 8.8) *	0.60 (0.20, 1.0) ^†^
Femurs	3	5.9 (4.0, 7.8)	4.6 (3.3, 5.8) *	3.0 (1.4, 4.6) ^†^	1.6 (0.9, 2.4)	1.9 (1.3, 2.5)	0.49 (0.10, 0.88) ^†^
Pelvic ilium	3	13.1 (11.3, 15.0)	9.8 (8.6, 11.0) *	9.1 (7.6, 10.7)	5.7 (4.5, 6.9)	7.3 (6.2, 8.3) *	2.9 (2.0, 3.8) ^†^
Sacrum	3	18.9 (17.0, 20.8)	14.0 (12.4, 15.6) *	14.6 (13.0, 16.2)	14.9 (13.2, 16.7)	12.2 (10.7, 13.8) *	10.1 (8.8, 11.6) ^†^
Vertebral bodies	8	26.6 (24.1, 29.1)	21.9 (19.6, 24.3) *	21.5 (19.1, 24.0)	21.5 (19.6, 23.4)	16.2 (14.3, 18.1) *	15.9 (14.2, 17.6)

Abbreviations: EUD = Equivalent Uniform Dose; 3D = 3D conformal photons; VMAT = Volumetric Modulated Arc Therapy photons; PBS = pencil-beam scanning protons.

**Table 3 cancers-16-00784-t003:** Secondary cancer Excess Absolute Risk (EAR) per 10,000 person–years and Lifetime Attributable Risk (LAR) projected to age 70 and averaged over 10 patients, with 95% confidence intervals for the mean given in parentheses. All PBS proton EAR and LAR values were significantly lower than 3D photon values (*p* < 0.05). Values that are significantly different between 3D and VMAT are labeled with * while values that are significantly different between VMAT and PBS are labeled with ^†^.

Organ	3D EAR	VMAT EAR	PBS EAR	3D LAR (%)	VMAT LAR (%)	PBS LAR (%)
Bladder	2.4 (2.2, 2.6)	3.0 (2.5, 3.6)	0.86 (0.66, 1.05) ^†^	0.34 (0.29, 0.38)	0.42 (0.34, 0.51)	0.12 (0.09, 0.15) ^†^
Large bowel	3.0 (2.7, 3.3)	1.8 (1.6, 1.9) *	0.41 (0.26, 0.55) ^†^	0.22 (0.20, 0.25)	0.13 (0.12, 0.14) *	0.03 (0.02, 0.04) ^†^
Small bowel	2.6 (2.4, 2.8)	1.4 (1.2, 1.7) *	0.59 (0.48, 0.71) ^†^	0.19 (0.17, 0.21)	0.10 (0.09, 0.12) *	0.04 (0.03, 0.05) ^†^
Spinal cord	8.2 (7.7, 8.7)	5.9 (5.2, 6.6) *	6.1 (5.6, 6.7)	1.15 (1.03, 1.27)	0.82 (0.70, 0.95) *	0.85 (0.76, 0.95)
Kidneys	3.5 (3.2, 3.8)	3.4 (3.1, 3.8)	0.81 (0.64, 0.98) ^†^	0.26 (0.23, 0.28)	0.25 (0.23, 0.28)	0.06 (0.05, 0.07) ^†^
Liver	1.17 (1.04, 1.30)	1.70 (1.62, 1.79) *	0.10 (0.05, 0.15) ^†^	0.13 (0.11, 0.15)	0.19 (0.18, 0.21) *	0.012 (0.005, 0.018) ^†^
Pancreas	1.84 (1.66, 2.01)	1.17 (0.98, 1.36) *	1.14 (1.01, 1.28)	0.14 (0.12, 0.15)	0.09 (0.07, 0.10) *	0.08 (0.07, 0.10)
Rectum	3.2 (1.9, 4.6)	4.7 (3.4, 6.0) *	2.0 (0.9, 3.0) ^†^	0.46 (0.25, 0.66)	0.67 (0.46, 0.87) *	0.28 (0.12, 0.44) ^†^
Stomach	3.2 (3.0, 3.3)	3.6 (3.4, 3.7) *	0.32 (0.18, 0.46) ^†^	0.49 (0.43, 0.54)	0.55 (0.48, 0.61) *	0.05 (0.02, 0.07) ^†^
Total	29.1	26.7	12.3	3.4%	3.2%	1.52%

Abbreviations: 3D = 3D conformal photons; VMAT = Volumetric Modulated Arc Therapy photons; PBS = pencil-beam scanning protons. According to the 2019 SEER (Surveillance Epidemiology and End Results) [1], the general-population incidence of cancers (absolute risk) per 10,000 person–years for the eight organs included in the table are as follows: bladder 1.8, large bowel 2.6, small bowel 0.25, kidneys 1.7, liver 0.9, pancreas 1.3, rectum 1.1, and stomach 0.7. Parentheses contain 95% confidence intervals, which are taken from the average of the model estimates over 10 patients and do not reflect inherent inaccuracies of the models themselves.

## Data Availability

Patient data are not available under the terms of the IRB for minimal risk. Secondary cancer analysis code is stored in an institutional repository and will be shared upon request to the corresponding author.

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
