# Peer review of "Radiation Therapy for Stage IIA/B Seminoma: Modeling Secondary Cancer Risk for Protons and VMAT versus 3D Photons"

_cancers, 2024, doi:10.3390/cancers16040784_

Round 1
Reviewer 1 Report
Comments and Suggestions for Authors
I have reviewed the manuscript titled "Radiation Therapy for Stage IIA/B Seminoma: Modeling the Risks of Secondary Cancers," several potential weaknesses and considerations are identified:
1. Retrospective Modeling Study: The study is retrospective and relies on modeling rather than direct patient outcomes. This approach may not capture the full complexity and variability of clinical scenarios.
2. Small Sample Size: The study is based on a small cohort of only ten patients. This limited sample size could affect the generalizability of the findings and may not adequately represent the broader patient population.
3. Variability in Patient Characteristics: There is considerable variability in patient ages and the anatomical locations of enlarged lymph nodes. This variability can introduce discrepancies in mean doses and Equivalent Uniform Doses (EUDs) to some organs, potentially affecting the study’s conclusions.
4. Assumption of Constant Proton Relative Biological Effectiveness (RBE): The study assumes a constant proton RBE of 1.1. However, RBE can vary based on physical and biological factors, and a higher RBE in some regions could impact dose and secondary cancer risks in organs-at-risk (OAR).
5. Lack of Prospective Data: The study suggests that prospective treatment protocols assessing outcomes and toxicity endpoints would be more effective in measuring the clinical impact of dosimetric improvements.
6. Exclusion of Certain Factors: The study does not consider the effects of lowdose scatter radiation or secondary neutrons outside the radiation field and the potential for secondary cancers in out-of-field organs.
7. Institutional Guidelines and Planning Techniques: The study acknowledges that results could be influenced by institutional guidelines and planning techniques, particularly for Volumetric Modulated Arc Therapy (VMAT).
8. Uncertainties in Model Parameters: There are acknowledged uncertainties in the model parameters used to estimate secondary cancer risks, which are not included in the secondary cancer estimates.
9. Limited Access and Availability of Proton Therapy: The study notes that access to proton therapy is still relatively limited in the United States, making It is a limited resource due to expense and availability.
10. Insurance Coverage and Treatment Policies: There is a discussion on insurance coverage for proton therapy and the categorization of testicular cancer in ASTRO Model Policies, suggesting a need for policy revision based on the study’s findings.
Author Response
- Thank you for the comment. We agree this is a limitation of the study and have included it in the discussion (lines 244-245).
- Thank you for the comment. We agree this is a limitation of the study and have included it in the discussion (lines 275-276).
- Thank you for the comment. We agree this is a limitation of the study and have included it in the discussion (lines 276-277).
- Thank you for the comment. We agree this is a limitation of the study and have included it in the discussion (lines 269-271).
- Thank you for the comment. We agree this is a limitation of the study and have included it in the discussion (lines 251-252).
- Thank you for the comment. We agree this is a limitation of the study and have included it in the discussion (lines 257-259).
- Thank you for the comment. We agree this is a limitation of the study and have included it in the discussion (lines 263-264).
- Thank you for the comment. We agree this is a limitation of the study and have included it in the discussion (lines 253-254).
- Thank you for the comment. We have included a discussion of proton availability in the conclusion (lines 289-290).
- Thank you for the comment. We have included a discussion of ASTRO model policies in the conclusion (lines 295-296).
Reviewer 2 Report
Comments and Suggestions for Authors
The risk of secondary cancers following radiotherapy is largely stochastic and the proposed model has not been validated. This risk cannot be quantified, as these studies cannot be done. As you mentioned in your text, the formula you are using is based on nuclear warfare victims. So this should be considered when making claims that proton beam is better compared to VMAT.
This is a small retrospective dosimetric analysis. The hypothesis is interesting, but the data presented can hardly confirm the benefit of proton beam.
Author Response
Thank you for the comments. We agree that the large uncertainties in secondary cancer risk models is a limitation and have included that in the discussion (lines 253-254). However, since this study is a comparison of secondary cancer risks based on dosimetric changes from different modalities, we do not expect the comparisons to be significantly impacted by modifications to model parameters (lines 255-257).
While this study is small and retrospective, it supports several other retrospective modeling studies and one prospective study which came to the same conclusion, that proton beam therapy is beneficial for treatment of seminoma. See Ref 7 (Efstathiou et al.), Ref 8 (Simone et al.), Ref 9 (Choo et al.), Ref 10 (Pasalic et al.), and Ref 11 (Maxwell et al.). Because this subject is understudied in the literature, we believe even this small retrospective study holds value to add to the published evidence. We agree that the clinical impact of dosimetric improvements would be best measured by a prospective treatment protocol that assesses outcomes and toxicity endpoints (lines 251-252).
Reviewer 3 Report
Comments and Suggestions for Authors
Summary
The article presents results of the modelling risks of secondary malignancies after testicular cancer radiotherapy. The risks of 3D-CRT, VMAT and PBS proton therapy are compared. The basic hypothesis of substantial advantage of PBS was verified as well as absence of advantage comparing 3D-CRT vers. VMAT. The expected long-term effect, far beyond the scope of consistent follow up and observation, is assessed on the basis of exact dosimetry results and relevant risk modelling. The method of both dosimetry comparison and risk modelling has been previously employed for the same subject it is duly cited in references. The article presents an effective option how to prove the dosimetry may result in a real clinical advantage.
General concept
The dosimetry of radiotherapy for seminoma has been assessed in several papers before, using the same technology of dosimetry comparison and risk modelling. This article is unique in modelling for stage IIA-IIB separately and pencil beam scanning proton treatment. Despite the topic and methodology of assessment have been published, the more focused insight is innovative. The results, not surprising, are designated to contribute to a wider availability of proton treatment to seminoma management. The results of modelling dosimetry vers. long-term clinical advantage in 10 patients may not be so influential. Anyway, also the equivocal results comparing 3D-CRT vers. VMAT may be influential proving there are limited options to decrease the long-term risks within the photon radiotherapy.
Review, specific comments
The dosimetric data are well arranged and comprehensive and dose calculation process is described in detail. The dosimetry is indicated in few examples of isodose plans, including VMAT technology despite it has not been employed clinically in any of the presented cases. However, assessment of VMAT planning is definitely beneficial. The results of secondary malignancy risk modelling are comprehensive as well and presented in clear lucid tables.
The text is clear and well structured according to the nature of the topic.
The presented results are unique for stage IIA, IIB seminoma and for PBS. This may be the first study modelling the risk of secondary malignancy in this group of patients.
The conclusions are plausible, and support the option of proton radiotherapy. The article may be strongly influential in this aspect. However, the topic is rather specialized concerning small community of radiation oncologists and small group of patients.
All significant statements and assumptions related directly to the topic are supported by citations. 16 of 30 citations are older than 10 years. However, the nature of the topic, especially slower development of risk assessment models, is relevant. The citation age does not do influence the quality of the article anyway.
There are no ethic aspects. The subject is dosimetry and modelling. The data are completely anonymized.
There are several specific comments:
· Page 2, row 67-69. I do not find necessary to comment the Bragg peak nature. It may be well known to all readers involved in proton treatment and considering its dosimetry.
· Figure 1, legend. It is recommendable to add labels to each of 6 pictures and present them in the legend. The table presenting colors of delineation (inserted between picture 4 and 5) is more confusing than explanatory. The are similar colors of isodose curves and color-wash.
· Table 2. The “a” parameter should be mentioned in the legend.
· Table 2, Table 3. The significance of EUD, EAR, LAR differences is presented. Compared 3D to VMAT and VMAT to PBS. Why not to present significance of 3D compared to PBS? Especially if 3D and PBS is more commonly used (e.g. in the presented patient group).
· Table 2, Table 3. The values of confidence intervals are presented in parenthesis. It is recommendable to use hyphen instead of comma between.
· Page 8, row 296-298. It is recommended to revisit ASTRO model policy since it allocates seminoma to group 2, which specifies the condition: “……coverage is recommended only if sparing of the surrounding healthy tissue cannot be adequately achieved with photon-based radiotherapy…..” This modelling study presenting convincing results proves exactly this. Indeed, the condition is met. Are You convinced the presented results are enough to move seminoma to group 1?
Conclusion
The results of dosimetric and long-term risk modelling study are presented. Innovative in focus on stage IIA-IIB seminoma and PBS proton therapy. The results clearly confirm the established hypotheses: Presentation is comprehensive and well structured. It may impact the indication of proton radiotherapy in a relatively narrow range of patients.
The article is recommended to be accepted after minor revisions.
Author Response
Thank you for the feedback and we appreciate the positive comments.
- Lines 67-68: We agree that all readers familiar with radiation oncology understand the Bragg peak. However, we hope that this article may also be of interest to medical oncologists researching the role of radiotherapy in the management of seminoma cancer, so we would like to leave this statement in place.
- Thank you for this feedback. We have made the following modifications to Figure 1:
- Removed the table presenting colors of delineated contours.
- Added legends for the six panels describing the plan.
- Thank you for this comment. We have added the following sentence to the caption of Table 2: “The “a” value is an organ-specific parameter used to compute the EUD from a nonuniform dose distribution.”
- Thank you for this comment. The EUD, Mean Dose, EAR, and LAR were also compared between 3D and PBS, and were found to be significantly improved by PBS for every organ. To reduce clutter in the table, the following statements were added to the captions rather than using a marker next to each value to indicate significance: “All PBS proton EUD and mean dose values were significantly lower than 3D photon values (p < 0.05)” (lines 191-192) and “All PBS proton EAR and LAR values were significantly lower than 3D photon values (p < 0.05)” (lines 219-220).
- Thank you for this comment. The APA Style Publication Manual recommends using a comma between the lower limit and upper limit of confidence intervals rather than a hyphen.
- Thank you for this feedback. We agree that the results of this study alone are not enough evidence to move seminoma from group 2 to group 1. However, this study adds to the increasing body of literature which predicts or demonstrates a benefit of protons for seminoma treatment [see Ref 7 (Efstathiou et al.), Ref 8 (Simone et al.), Ref 9 (Choo et al.), Ref 10 (Pasalic et al.), and Ref 11 (Maxwell et al.)]. The recommendation to reconsider the classification of seminoma as group 2 is given in the context of the larger body of evidence, not based solely on this study.
Reviewer 4 Report
Comments and Suggestions for Authors
In this article the authors study radiation therapy for stage IIA/B seminoma by modeling secondary cancer risk for protons and VMAT versus 3D photons.
General comment: Although the study has several limitations like for example the number of patients that are followed, the results of this article are valuable and important to share. The manuscript is well written. Therefore, this reviewer supports its publication.
Minor point: The font type is not uniform across the article (e.g. Table 1).
Author Response
Thank you for the feedback and we appreciate the positive comments. We have checked the font type across the manuscript.
Reviewer 5 Report
Comments and Suggestions for Authors
The authors should be congratulated for their work. The study aimed to assess the differences in secondary cancer risk for stage II seminoma patients after three different types of radiation therapy (RT). Stage II seminoma is not a life-threatening disease, compared to stage III or IV seminoma or other germ cell tumors (PMID= 37640983, 37558516, 35318754). However, the methodology presented several lacks that should be addressed in a major revision.
- Firstly, the sample size is too small and highly heterogeneous. What is the median age of the two groups? The effects of RT should be weighted for the comorbidities of the two groups that I missed in this paper. Could the authors provide more details? When the other cause mortality is not adjusted for, the cancer-specific mortality of these patients should be misleading to be interpreted. This represents an inherent limitation.
- What were the selection criteria for RT? Why did the patients choose the RT rather than chemo? Consent provided to these patients should be included in the supplementary materials. Did they receive also chemotherapy before RT or RT is the only adjuvant treatment?
Despite the authors' commitment, my suggestions will highlight the strong points of the paper. It is nebulous to interpret some data, without information of the population studied, above all if the sample size is limited. The results derived should be interpreted with great caution.
Author Response
Thank you for the feedback and we appreciate the positive comments.
- We agree that one limitation of this study is the small and heterogenous patient population, as discussed in lines 275-277. This is due to the rare nature of CS2 seminoma, with only ~10% of new cases diagnosed at this stage (lines 45-46).
- The median age of the CS2A group is 41 while the median age of the CS2B group is 38, as can be determined from Table 1.
- Because this is a retrospective planning study, the actual patient outcomes from treatment were not considered part of the results (lines 244-245). For this reason, we did not include information about patient comorbidities or control for those in the results. No actual mortalities are reported in the results, either cancer-specific or other cause. Instead, the results of this study are the calculated risks of developing a secondary cancer from the dose to normal organs received during a course of radiotherapy delivered by protons or photons (3D or VMAT).
- Because this is a retrospective study, not a prospective study, there were no selection criteria for receiving RT and no information included on why patients chose to receive RT.
- Patient consent was waived by the IRB due to this being a retrospective study which is limited to the use of health/medical information and study results are published without patient identifiers.
- We did not include information about chemotherapy treatments as part of this study as it was not relevant to the comparison of secondary cancer risk from different RT treatment modalities.
- Thank you for the comments. Because this subject is understudied in the literature, we believe even this small retrospective study holds value to add to the published evidence.
Round 2
Reviewer 2 Report
Comments and Suggestions for Authors
Well presented dosimetrical study.
Useful comparison of protons to IMRT.
Reviewer 5 Report
Comments and Suggestions for Authors
The authors properly addressed my comments.